# Nutritionist-Guided Video Intervention Improves Adherence to Mediterranean Diet and Reduces the Rate of Gestational Diabetes Mellitus: A Randomized Clinical Trial

**DOI:** 10.3390/nu17223533

**Published:** 2025-11-12

**Authors:** Rocío Martín-O’Connor, Ana M. Ramos-Levi, Ricardo Saviron-Cornudella, Bricia López-Plaza, Angélica Larrad-Sainz, Ana Barabash, Clara Marcuello-Foncillas, Inés Jiménez-Varas, Angel Diaz-Perez, Paz de Miguel, Miguel A. Rubio-Herrera, Pilar Matía-Martín, Alfonso L. Calle-Pascual

**Affiliations:** 1Departamento de Endocrinología y Nutrición, Hospital Clínico San Carlos, Instituto de Investigación Sanitaria San Carlos (IdISSC), 28040 Madrid, Spain; rocio@oconnor.es (R.M.-O.); ana_ramoslevi@hotmail.com (A.M.R.-L.); blopezpl@ucm.es (B.L.-P.); a.larrad47@gmail.com (A.L.-S.); ana.barabash@salud.madrid.org (A.B.); claramarcuello@gmail.com (C.M.-F.); i.jimenez.varas@gmail.com (I.J.-V.); joseangel.diaz@salud.madrid.org (A.D.-P.); pazdemiguelnovoa@gmail.com (P.d.M.); marubioh@gmail.com (M.A.R.-H.); 2Facultad de Medicina, Departamento de Medicina II, Universidad Complutense de Madrid, 28040 Madrid, Spain; 3Centro de Investigación Biomédica en Red de Diabetes y Enfermedades Metabólicas Asociadas (CIBERDEM), 28029 Madrid, Spain; 4Departamento de Obstetricia y Ginecologia, Hospital Clínico San Carlos, Instituto de Investigación Sanitaria San Carlos (IdISSC), 28040 Madrid, Spain; ricardo.saviron@salud.madrid.org

**Keywords:** gestational diabetes mellitus, nutritional intervention, telemedicine, Mediterranean diet

## Abstract

**Aims:** Gestational diabetes mellitus (GDM) represents an increasing global challenge. Mediterranean diet interventions have proven benefits, but their implementation is limited by the absence of nutritionists in many public health systems. This study aimed to evaluate whether a video intervention guided by a nutritionist could increase compliance to Mediterranean diet and reduce the incidence of GDM and adverse maternal–neonatal outcomes. **Methods:** In this randomized controlled trial, 1750 consecutive pregnant women were allocated (1:1) to standard care (verbal, printed advice) or to a video designed by a nutritionist promoting a Mediterranean and physical activity. The primary outcome was the incidence of GDM; secondary outcomes included other pregnancy-related complications. Dietary adherence was assessed using the 14-item Mediterranean Diet Adherence Screener (MEDAS) score. **Results:** The experimental group increased their MEDAS score from baseline to GDM screening (mean difference (95% CI) 0.41 (0.23; 0.60); *p* < 0.001), mainly through greater extra virgin olive oil and nut intake and lower consumption of juices and confectionery. GDM incidence declined from 25.1% to 20.7% (*p* = 0.025), with significant reductions in gestational hypertension, episiotomy and neonatal intensive care unit admissions. **Conclusions:** Nutritionist-guided video intervention improves adherence to Mediterranean diet and reduces GDM incidence and adverse outcomes. This low-cost, scalable approach may help overcome structural limitations in public health systems.

## 1. Introduction

Gestational diabetes mellitus (GDM) is the most prevalent metabolic disorder during pregnancy. According to the International Diabetes Federation, it affects around 14% of pregnancies worldwide [1,2] and up to an alarming 37.6% of pregnancies in Spain [3], although this prevalence varies according to the heterogeneity of the diagnostic criteria and the population studied [2,4]. These data reflect a worrying trend that places GDM as one of the main contemporary obstetric challenges that is increasing in parallel with obesity and type 2 diabetes [1].

Nutritional treatment represents the cornerstone and first-line approach for the management of GDM, and may be sufficient in 70–85% of cases [5,6]. In this regard, a recent meta-analysis showed that nutritional interventions can significantly reduce the risk of developing GDM by 18–27% and entail benefits in the control of maternal and neonatal metabolic outcomes when implemented in the first weeks of pregnancy [7,8,9,10].

In our Healthcare Area, the adoption of International Association of Diabetes and Pregnancy Study Groups (IADPSG) criteria raised GDM prevalence from 10.6% to 35.5% [11]. This prompted the St. Carlos GDM Prevention Study, showing that an early Mediterranean diet, prescribed before 12 weeks and supervised by a clinical nutritionist, reduced GDM incidence and improved maternal–fetal outcomes [10]. These findings were later confirmed in routine practice through a prospective study, lowering GDM prevalence to 13.9% [12] in line with the World Health Organization Global Diabetes Compact promoting universal, cost-effective interventions [13].

However, most successful programs rely on face-to-face counseling by clinical nutritionists, a resource largely absent in many public health systems. Digital health strategies, including apps, tele-education, and remote monitoring, have emerged as promising alternatives [14,15], but their scalability and feasibility remain uneven.

In this context, video-based education represents a pragmatic and innovative approach: it delivers standardized, evidence-based counseling at minimal cost, can be universally implemented during the first prenatal visit, and avoids the high personnel requirements of individualized interventions.

This study evaluates the impact of an early, nutritionist-designed video on GDM incidence and maternal–neonatal outcomes, with a focus on its feasibility as a scalable public health tool. Secondary objectives include the evaluation of its effects on maternal–neonatal outcomes and determining its feasibility as a scalable public health intervention.

## 2. Materials and Methods

### 2.1. Study Design

This was a randomized interventional clinical trial (RCT), which was conducted in accordance with the Consolidated Standards of Reporting Trials (CONSORT) 2025 guidelines for reporting randomized trials. The protocol was developed following the principles of the Declaration of Helsinki and was approved by the Research Ethics Committee of the Hospital Clínico San Carlos, Madrid, Spain (Promoter Code: ESTUDIO TELE_MED_DIET; Internal Code: 23/616-E).

The trial was registered at ISRCTN (ISRCTN20122349). All participants provided written informed consent before enrollment.

### 2.2. Setting and Population

The study was conducted at Hospital Clínico San Carlos, a tertiary academic medical center in Madrid, Spain, which serves a diverse urban population. Eligible participants were pregnant women who attended consecutively their first prenatal visit before 16th gestational week (GW) (to ensure at least 12 weeks of exposure to the intervention), with a single pregnancy, 18 years old or more and the ability to read, understand, and consent to the intervention in Spanish. Exclusion criteria were impaired fasting plasma glucose (>92 mg/dL) at the time of first assessment (weeks 8–12), multiple pregnancy, assisted reproduction or in vitro fertilization (IVF), nut allergy, and unavailability to give informed consent or to download the intervention video.

### 2.3. Randomization and Allocation

Randomization was stratified by weekday clinics, alternating schedule designed to ensure a pragmatic yet unbiased allocation of participants. Specifically, for one week, women attending prenatal visits on three predetermined weekdays (Monday, Wednesday, Friday) were assigned to the experimental group, while those attending on the remaining two days (Tuesday, Thursday) were assigned to the control group. The following week, the allocation was reversed (the same weekdays were assigned to the control group and the others to the intervention group). This alternating assignment by working days was implemented to ensure pragmatic randomization in the consecutive flow of patients, avoiding any subjective selection by researchers. This procedure maintained random patient flow within each clinic day and avoided any subjective selection by investigators. Allocation was implemented by obstetricians during the first prenatal visit (<16 GW) by providing a unique Quick-Response (QR) code to experimental group participants. It should be noted that a considerable number of women in the experimental group did not download the video, since randomization did not account for whether participants had mobile devices or applications compatible with QR code scanning. This design, however, avoided the potential bias of selecting women with greater access to digital technologies. Data analysts remained blind to group allocation throughout the statistical analysis. Flow-chart of participant recruitment, allocation, follow-up, and analysis is presented in Figure 1.

### 2.4. Intervention and Follow-Up

At the first obstetric visit, women in the experimental group received a QR code linking to an 8 min video narrated by a clinical nutritionist. The video, recorded in Spanish and including Spanish subtitles, is freely available on the YouTube platform and designed to ensure universal accessibility. Translation into other languages, including English, is planned to facilitate future multicenter implementation. The video contained pregnancy-adapted recommendations on:-*Mediterranean diet adherence*: a minimum of 3 servings per day of dairy products. Fruit and vegetables should be eaten daily, with at least 12 servings of each of these groups per week. Whole fruit should be prioritized over juices. Extra virgin olive oil (EVOO) should be used as the main fat. No less than four tablespoons per day are recommended, including both raw for dressing dishes and homemade vegetable stir-fries with this olive oil. This will avoid commercial sauces. A handful of nuts is recommended every day, mainly pistachios. Pregnant women should be encouraged to choose whole foods instead of refined ones (>5 days per week). The recommendation for legumes consumption is 3 servings per week and the recommendation for consumption of oily fish is, at least, three servings per week, including canned fish, mainly small ones. Avoidance of processed foods (<2 days per week) was recommended, and instead consume white meats such as chicken, turkey, or rabbit over red meats. Commercial confectionery and processed snacks should be avoided (<2 days per week). Drinking regular water should be prioritized over sugary drinks such as soft drinks, milkshakes, and juices, which should be avoided (<2 portions per week). In addition, the video provided several meal ideas with images to make it more appealing. The aim was to increase adherence and simplify the intervention recommendations.-*Physical activity recommendations* included moderate strength training adapted to pregnancy, an active lifestyle with ≥1 h/day walking, stair climbing, and avoiding prolonged sitting.

The video clarified that recommendations were general and should be adapted according to obstetrician instructions. Participants had unlimited access and could view it ad libitum.

Control group participants received routine verbal advice and printed brochures with standard recommendations.

Intervention adherence was verified at 24–28 GW at the time of the oral glucose tolerance test using the 14-point Mediterranean Diet Adherence Screener (MEDAS) and Nutritional Score, along with the International Physical Activity Questionnaire (IPAQ) to assess physical activity. Compliance was defined by watching the complete video at least one time before 24–28 GW. Verification was performed by the nutritionist through structured questioning during the follow-up assessment.

### 2.5. Clinical Outcomes

The primary outcome was the incidence of GDM diagnosed at 24–28 GW according to the (IADPSG criteria: fasting plasma glucose ≥ 92 mg/dL, 1 h ≥ 180 mg/dL, or 2 h ≥ 153 mg/dL after a 75 g oral glucose tolerance test, with one or more abnormal values [16].

Secondary outcomes included maternal weight gain, hypertensive disorders of pregnancy (systolic blood pressure ≥ 140 mmHg and/or diastolic ≥ 90 mmHg after 20 GW in previously normotensive women), preeclampsia (hypertension after 20 GW with proteinuria ≥ 300 mg/24 h or equivalent evidence of maternal organ dysfunction) [17]. Delivery was classified as vaginal spontaneous, instrumental, or cesarean section. Preterm birth was defined as delivery before GW 37. Birthweight was recorded in grams, and newborns were classified as small for gestational age (SGA, <10th percentile) or large for gestational age (LGA, >90th percentile) according to INTERGROWTH-21st standards [18]. Neonatal hypoglycemia was defined as plasma glucose <40 mg/dL within the first 24 h of life. Admission to the neonatal intensive care unit (NICU) was considered for any indication regardless of length of stay [19].

### 2.6. Data Collection and Variables

Baseline and follow-up data were obtained from electronic medical records and standardized questionnaires administered at the first prenatal visit and at 24–28 GW by a clinical nutritionist:-Clinical history: personal and family history of diabetes, hypertension, dyslipidemia, obesity, obstetric history, medications, supplements, smoking status.-Anthropometric and clinical data: pregestational weight and height, body mass index (BMI), blood pressure.-Biochemical parameters: Blood and urine samples were obtained after an overnight fast of at least 8 h. Fasting plasma glucose (mmol/L) was determined in serum by the glucose–hexokinase method using an AU5800 analyzer (Beckman Coulter Diagnostics, Brea, CA, USA). Glycated hemoglobin (HbA1c, %) was measured by ion-exchange high-performance liquid chromatography (HPLC) on a Tosoh G8 analyzer (Tosoh Co., Tokyo, Japan). The method is standardized against the International Federation of Clinical Chemistry, with an imprecision of 1.23% for 32.23 mmol/mol (5.1% NGSP) and 1.36% for 85.24 mmol/mol (10% NGSP). Fasting serum insulin (μIU/mL) was quantified by chemiluminescence immunoassay (IMMULITE 2000 Xpi, Siemens Healthcare Diagnostics, Munich, Germany). Inter-assay CVs were 6.3% at 11 μIU/mL and 5.9% at 21 μIU/mL. Insulin resistance was estimated using the homeostasis model assessment (HOMA-IR): glucose (mmol/L) × insulin (μIU/mL)/22.7. All laboratory methods were subject to monthly external quality assurance by the Spanish Society of Clinical Chemistry (SEQC), which also performed regular method validation and review.-Dietary adherence using adapted 14-point MEDAS (12-point score; range 0–12).-Physical activity was assessed using the IPAQ, which records time spent walking, stair climbing, aerobic activities and moderate-intensity exercise. Daily walking time (minutes/week) and stair climbing (number of floors per day, >5 days/week) were specifically recorded, together with the frequency of aerobic and moderate-intensity exercise.-To assess patient satisfaction, at the end of the dietary questionnaires, participants in the experimental group were asked whether the information provided in the video had been clear and whether they found it useful. This exploratory assessment aimed to collect feedback for the potential improvement of the educational video.

### 2.7. Sample Size Calculation

Sample size was calculated primarily with the aim of estimating the rate of GDM between 24 and 28 weeks of gestation. A preliminary analysis compared the first 100 women who received the video intervention (20 cases of GDM, 20%) with the previous 100 women who received standard recommendations (25 cases of GDM, 25%). Assuming a baseline prevalence of 25% in the control group, we aimed to detect a 5% absolute reduction in the experimental group (from 25% to 20%). For a two-sided comparison of two proportions (α = 0.05, power = 80%), approximately 628 women were required per group. Allowing for a 5% dropout rate, the final sample size increased to 662 women per group (1324 in total). Ultimately, 1750 consecutive women were enrolled to ensure adequate statistical power. The power of the study to detect a 25% difference is approximately 95%. If the actual reduction were only 20%, the study’s power would be 81–83% to detect this difference.

### 2.8. Statistical Analysis

Data are expressed as mean and standard deviation (SD) for continuous variables and as absolute numbers and percentages for categorical variables. Categorical variables were analyzed using the χ^2^ test or Fisher’s exact test, and continuous variables using Student’s *t*-test or Mann–Whitney U test, as appropriate. Relative risk (RR) and 95% confidence intervals (CI) were calculated for categorical outcomes. Analyses followed the intention-to-treat principle, with additional per-protocol analysis based on adherence criteria. Missing data were handled through complete-case analysis. Multivariate logistic regression analyses were performed adjusting for age, pre-pregnancy BMI, parity, and ethnicity for the primary outcomes (GDM rate).

## 3. Results

A total of 1750 women were assessed for eligibility (experimental group: *N* = 823; control group: *N* = 823). After exclusion criteria, 1529 women were included in the analysis (experimental group: *N* = 715; control group: *N* = 814). Baseline demographic and clinical characteristics of women included were well balanced between groups (Table 1). Although statistically significant differences were observed in ethnicity and blood pressure at baseline, these reflected the real demographic composition of our catchment population rather than selection bias. Furthermore, differences in sBP and dBP are opposite in each group, so the difference in mean BP is not significant. Moreover, participants were asked about their length of residence in Spain to account for acculturation effects, and the protective impact of Mediterranean diet adherence has been consistently demonstrated across diverse ethnic groups.

From baseline to GDM screening at 24–28 weeks of gestation, the experimental group significantly increased their mean MEDAS score (mean difference [95% CI]: +0.41 [0.23; 0.60], *p* < 0.001), whereas the control group showed a decrease (−0.22 [−0.30; −0.14], *p* < 0.001). Between-group differences were statistically significant (*p* < 0.001 for trends) (Table 2).

The improvement in the MEDAS score among intervention participants was primarily explained by a significant increase in EVOO consumption (>40 mL/day) and weekly servings of pistachios/nuts, alongside a significant reduction in the intake of fruit juices and sugar-drinks and in the consumption of commercial confectionery products. No significant differences were observed in other major food components or in daily walking time between groups, although stair climbing frequency improved modestly in the intervention arm. Table 3 shows adverse outcomes by groups.

As shown in Table 3, the incidence of GDM (primary endpoint) was significantly lower in the intervention group compared with the control group (20.7% vs. 25.1%, *p* = 0.025). Among secondary maternal outcomes, the intervention group had a lower rate of pregnancy-induced hypertension (2.6% vs. 5.6%, *p* = 0.004), a lower rate of episiotomy (14.4% vs. 19.0%, *p* = 0.015), and lower admission rate in NICU (5.9% vs. 9.7%, *p* = 0.008). No significant differences were observed for preeclampsia, cesarean section rate, preterm birth, or neonatal anthropometric measures. No side effects related to the intervention were reported.

86 of the 823 participants (10.45%) experienced technical problems that prevented them from accessing the video. However, the 715 participants who downloaded the video (100%) considered the information supplied clear. Before GDM screening, 528 of 715 women (73.85%) viewed it more than once. Additionally, 146 of them (20.4%) reported lifestyle changes while 569 (79.6%) confirmed or reinforced existing habits.

## 4. Discussion

In this RCT, a video which promotes nutritional intervention, explicitly designed and guided by a clinical nutritionist, significantly improved the adherence to Mediterranean diet and entails a reduction in the incidence of GDM and several adverse maternal and neonatal outcomes, compared with routine clinical practice. The absolute reduction of 5% in the incidence of GDM implies an estimated number needed to treat (NNT) of 23 women to prevent one case, which represents a clinically relevant impact at the population level given the simplicity and low cost of the intervention. If universally applied, approximately 1 in 23 cases of GDM could be prevented, with positive repercussions on maternal–fetal morbidity and healthcare system costs. Although the absolute risk reduction (ARR) observed in our study could theoretically be used to estimate the NNT and its confidence interval, this parameter might not be appropriate for direct interpretation, as the intervention was not intended to replace standard face-to-face dietary counseling but to enhance adherence to the Mediterranean diet during pregnancy. Therefore, the ARR and relative risk reduction were considered more meaningful indicators of clinical relevance.

From a cost-effectiveness perspective, previous studies have estimated that preventing one case of GDM could save approximately EUR 818 per pregnancy in short-term healthcare costs, suggesting that even modest reductions in incidence may have a considerable impact when extrapolated to large populations.

These findings reinforce the notion that the effectiveness of digital resources is enhanced when their content is grounded in evidence-based nutritional counseling and delivered under the expertise of professionals in nutrition. Previous systematic reviews have shown that digital health tools, including tele-education, can improve glycemic control and some obstetric outcomes in women with GDM, but their effectiveness is greatest when professional guidance is integrated, suggesting that such tools should complement, rather than replace, individualized follow-up [14,15].

Our trial adds to the growing body of evidence that early nutritional interventions can modify the risk of GDM. Specifically, we demonstrate that a video intervention designed by a clinical nutritionistas supplied in the current study [20]—rather than generic recommendations—can strengthen adherence to the Mediterranean diet, a dietary pattern consistently associated with a 30–35% reduction in GDM incidence in meta-analyses [1,2,21,22,23]. Its nutrient density, low glycemic index, and anti-inflammatory properties are plausible mechanisms underlying these benefits [24,25]. However, the absolute reduction of 5% observed in our study, though clinically meaningful, is more modest than the reductions achieved in trials involving face-to-face counseling [10,11,12,13,14,15,21,22]. This suggests that while digital interventions offer important advantages in scalability and accessibility, they may lack the degree of personalization and reinforcement that individualized professional support provides.

The potential public health implications, even if the absolute reductions in GDM risk are modest, could translate into a significant number of cases prevented at the population level. This would result in benefits such as reduced need for insulin treatment, fewer hypertensive disorders of pregnancy, and fewer admissions to neonatal intensive care. Nonetheless, the scalability of digital tools must be balanced against the recognition that they should complement, not substitute, individualized specialist care, particularly in women with high metabolic risk.

A major strength of this intervention lies in its scalability, simplicity, and feasibility for large-scale implementation. The video developed can be disseminated universally at minimal additional cost, integrated into routine prenatal care, and standardized across diverse clinical settings. Compared with other digital modalities, such as mobile applications or continuous remote monitoring, video-based education requires minimal technological literacy, ensures message standardization, and can be implemented even in resource-limited contexts. The expansion of this model at the national level could be easily integrated into public prenatal care programs due to its low cost, reproducibility, and open-access availability on free platforms, which facilitates its adoption. Nevertheless, effective implementation would require ensuring digital accessibility, particularly in rural areas, and fostering the active participation of obstetricians, midwives, and other healthcare professionals as educational reinforcement agents.

Our study has some limitations. The use of dietary questionnaires entails potential recall and social desirability biases; although their administration by a qualified nutritionist likely reduced these risks, they cannot be completely excluded. The intervention was only available in Spanish, which may restrict its generalizability to multicultural settings. Technical difficulties that prevented the video from being downloaded affected 10.45% of the women assigned to the experimental group. This potential bias can be considered non-differential, as it was not related to documented clinical or socioeconomic characteristics, but rather to the limitations of electronic devices. However, we acknowledge that it may have underestimated the actual effect of the intervention by excluding women with lower digital literacy. In future implementations, we propose facilitating access through viewing in waiting rooms or multiplatform formats to address this issue and overcome this limitation. In addition, future research should incorporate validated biomarkers of diet adherence, develop hybrid models that combine universal digital education with individualized follow-up by nutritionists for women at higher metabolic risk, and evaluate both long-term postpartum outcomes and the cost-effectiveness of this strategy compared to alternative interventions.

## 5. Conclusions

In this RCT, an early telematic intervention promoting a Mediterranean diet enriched with extra virgin olive oil and nuts significantly improved adherence and helped to reduce the incidence of gestational diabetes mellitus and several adverse pregnancy outcomes, including gestational hypertension, episiotomy, and neonatal intensive care admissions. The intervention improved diet quality and adherence to Mediterranean dietary principles in a pragmatic, low-cost, and scalable way, making it a feasible strategy for public health systems with limited access to clinical nutritionists.

Future research should focus on hybrid models that combine universal telematic resources with tailored follow-up for high-risk women, multilingual adaptation, integration of objective biomarkers of adherence, and long-term cost-effectiveness evaluation.

## Figures and Tables

**Figure 1 nutrients-17-03533-f001:**
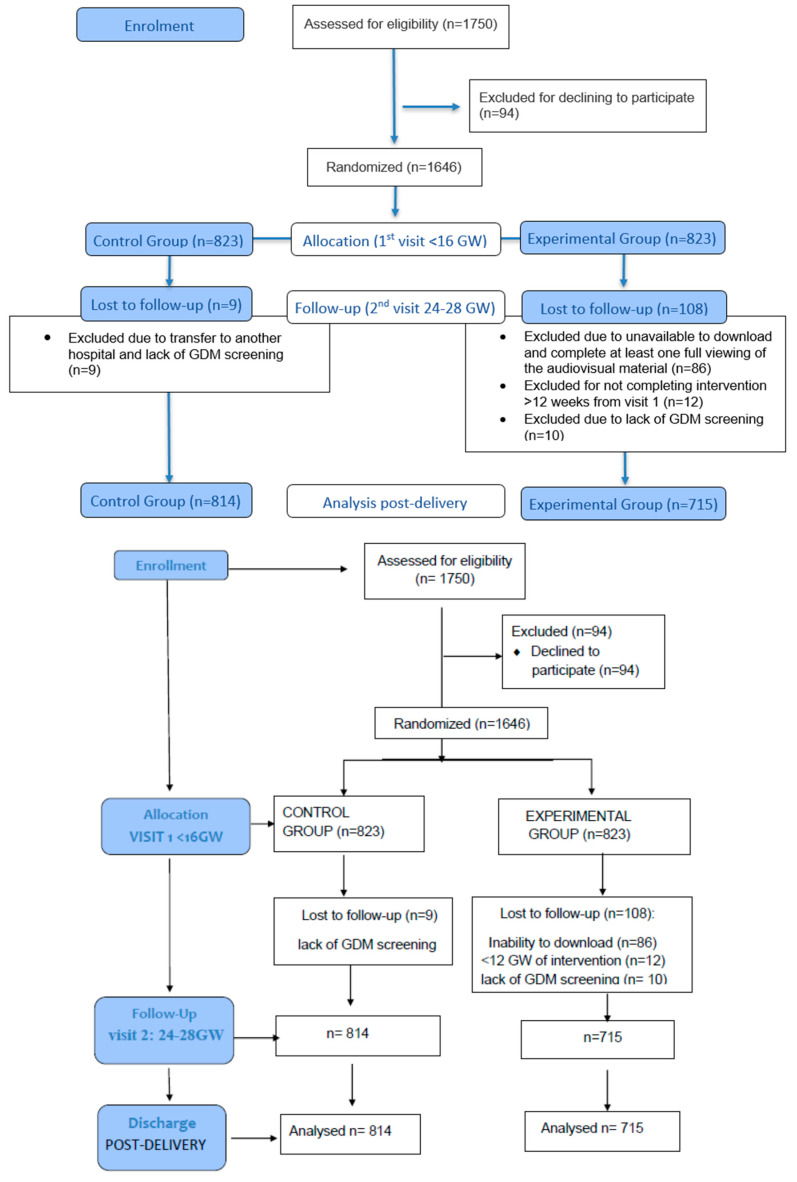
Flow chart of participants assessed.

**Table 1 nutrients-17-03533-t001:** Baseline characteristics of participants studied.

	Experimental Group	Control Group	*p* Value
*N*	715	814	
Age (years)	31.88 ± 5.56	32.41 ± 5.79	0.776
PP-BW (kg)	64.68 ± 13.09	64.50 ±12.78	0.279
PP-BMI (kg/m^2^)	24.8 ± 4.9	24.3 ± 4.7	0.048
sBP (mm Hg)	113.7 ± 11.9	116.3 ± 12.1	0.001
dBP (mm Hg)	76.2 ± 8.4	74.9 ± 8.9	0.005
FSG (mg/dL)	82.5 ± 5.8	81.7 ± 6.0	0.080
HbA1c %	5.2 ± 0.3	5.2 ± 0.3	0.114
Primiparous n (%)	278 (38.9)	352 (43.2)	0.301
Prior GDM	54 (7.6)	69 (8.5)	0.285
Prior Miscarriage	274 (38.3)	300 (36.9)	0.093
Never Smoker (%)	509 (71.2)	577 (71.1)	0.186
Ethnicity: Caucasian Latin American			0.000
	331 (46.3)	459 (56.4)	
	370 (51.7)	328 (40.3)	
Educational status: Low High	26 (3.6)	38 (4.6)	0.052
	433 (62.3)	506 (62.3)	
Occupation	534 (74.9)	627 (77.2)	0.309
MEDAS Score	7.3 ± 2.4	7.6 ± 2.3	0.169

Abbreviations: PP-BW, Prepregnancy body weight; PP-BMI, Prepregnancy; BMI, body mass index sBP, Systolic Blood Pressure; dBP, Diastolic Blood Pressure; FSG, Fasting Serum Glucose.

**Table 2 nutrients-17-03533-t002:** Results from questionnaires on lifestyle throughout pregnancy according to main targets of the video.

		At Baseline	24–28 GW	Mean Diferencies (95% CI)	*p* Value
Medas Score	Control Group	7.6 ± 2.3	7.2 ± 2.3	−0.22 (−0.30; −0.14)	0.001
	Experimental Group	7.3 ± 2.4	7.7 ± 2.3	0.41 (0.23; 0.60)	0.001
	*p* value	0.169	0.001	0.001	
Numbers of floors climbed per day. (>5 days a week)	Control Group	4.5 + 5.4	3.7 + 6.0	−0.8 (−0.5; −1)	0.001
	Experimental Group	4.7 + 6.0	4.8 + 5.6	0.6 (0.3; 0.8)	0.001
	*p* value	0.011	0.006	0.001	
Walking daily (>5 days) min per week	Control Group	546 + 412	462 + 342	−84 (−61; −106)	0.001
	Experimental Group	586 + 450	494 + 373	−93 (−66; −121)	0.001
	*p* value	0.072	0.098	0.075	
Pieces of fruit per week	Control Group	11 ± 9	16 ± 9	4 (4; 5)	0.001
	Experimental Group	10 ± 9	16 ± 9	5 (4; 5)	0.001
	*p* value	0.046	0.970	0.045	
EVOO > 40 mL /day N (%) or increase ml/day	Control Group	371 (45.7)	490 (60.3)	5.8 (−7.1; 10.3)	0.001
	Experimental Group	310 (43.4)	482 (67.4)	6.7 (−3.4; 5.7)	0.001
	*p* value	0.194	0.007	0.004	
Pistachio/Nuts (serving/weeks)	Control Group	2.3 ± 2.5	2.3 ± 2.6	−0.3 (−0.4; −0.2)	0.001
	Experimental Group	2.2 ± 2.5	3.0 ± 2.7	0.4 (−0.2; 0.8)	0.001
	*p* value	0.229	0.011	0.001	
Juice or sugar drinks (serving per week)	Control Group	3.0 ± 5.6	2.9 ± 4.5	−0.1 (−0.5; 1)	0.001
	Experimental Group	3.8 ± 6.4	2.1 ± 5.6	−0.7 (0.3; 1.1)	0.001
	*p* value	0.007	0.001	0.001	
Bakery/commercial confectionery servings per week	Control Group	3.1 ± 3.1	3.1 ± 2.8	−0.0 (−0.3; 0.1)	0.001
	Experimental Group	3.1 ± 3.8	2.9 ± 2.6	−0.2 (−0.4; −0.1)	0.001
	*p* value	0.095	0.095	0.045	
Maternal weight gain (Pregestational TO (kg))	Control Group	1.0 ± 2.4	5.4 ± 3.1	6.4 (−0.57; 8.77)	0.001
	Experimental Group	1.2 ± 2.7	4.9 ± 2.3	6.2 (0.73; 7.65)	0.001
	*p* value	0.186	0.444	0.081	

Abbreviations: Data mean + SD or n (%). EVOO, extra virgin olive oil. MEDAS Score, 14-point Mediterranean Diet Adherence Screener (MEDAS). *p*, denote differences between groups each time (*t*-test) and each group compared to baseline as mean of differences 95% CI (confidence interval).

**Table 3 nutrients-17-03533-t003:** Maternal and neonatal adverse outcomes by groups.

	Experimental Group	Control Group	*p*
*N*	715	814	
*Maternal outcomes*			
GDM	148 (20.7)	204 (25.1)	0.025
Insulin treatment	22 (14.9)	51(17.3)	0.060
Pregnancy-induced hypertension	17 (2.6)	44 (5.6)	0.004
Preeclampsia	14 (2.2)	30 (3.8)	0.051
Delivery			
Vaginal eutocic	474 (77.9)	590 (74.6)	0.189
Cesarean section	116 (19.1)	179 (22.8)	
Emergency-CS	80 (13.1)	110 (13.9)	0.418
Episiotomy	85 (14.4)	147 (19.0)	0.015
Perineal Trauma (any tears grade)	284 (482)	349 (44.9)	0.233
*Neonatal outcomes*			
Gestational Age at birth (weeks)	38.9 ± 1.6	39.0 ± 1.6	0.415
<37 GW	34 (5.6)	38 (5.0)	0.324
Birthweight (g)	3240 ± 434	3238 ± 481	0.800
Centile	50.6 ± 27.8	50.1 ± 27.4	0.774
Length (cm)	49.7 ± 2.4	49.7 ± 2.4	0.967
Centile	59.3 ± 28.4	58.4 ± 29.4	0.614
LGA > 90 percentile	50 (9.1)	61 (7.9)	0.240
SGA < 10 percentile	39 (7.1)	53 (6.8)	0.459
Apgar Score at 1 min	8.6 ± 1.7	8.6 ± 1.1	0.378
Apgar Score at 5 min	9.7 ± 1.0	9.8 ± 0.8	0.129
Hypoglycemia	20 (3.7)	39 (5.0)	0.152
NICU/observation	32 (5.9)	75 (9.7)	0.008

Abbreviations: Data are Mean + SD or number (%). Abbreviation GDM, Gestational Diabetes Mellitus; CS, C-section; GW, gestational weeks; LGA, large-for-gestational-age. SGA, small-for-gestational-age. NICU, Neonatal intensive care unit.

## Data Availability

The dataset generated and analyzed during the current study, together with the full video intervention, are available in the Zenodo repository at https://doi.org/10.5281/zenodo.16419364. Access is open and provided under a Creative Commons Attribution (CC-BY) license [20].

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
