# Peer review of "Nutritionist-Guided Video Intervention Improves Adherence to Mediterranean Diet and Reduces the Rate of Gestational Diabetes Mellitus: A Randomized Clinical Trial"

_nutrients, 2025, doi:10.3390/nu17223533_

Round 1
Reviewer 1 Report
Comments and Suggestions for Authors
To the authors
This manuscript examined the effects of video intervention on prevention of gestational diabetes mellitus (GDM). The authors provided a video containing advices about mediterranean diet and physical activity. The results showed the effectiveness. Although this manuscript has important results, there are some comments.
- The authors did not show maternal weight gain during pregnancy. I think that this is informative for readers.
- As a limitation, the authors mentioned the exclusion of some participants due to technical difficulties in downloading the video. I wonder if it caused some bias and made the results difficult to be interpreted. Please discuss it in more depth.
- I am interested in the effects of extending this method to the whole of Spain. I also recommend to consider the problems involved in expanding this throughout the country. Please discuss it.
- Abbreviations should be spelled out at the first time.
- There are some extra spaces in the main text.
- Table 1: Were history of GDM and miscarriage tested simultaneously? P value was shown as 0.004.
Author Response
Rev 1
We appreciate the constructive comments. Below we provide a detailed response, point by point, to each of the reviewers’ observations.
To the authors
This manuscript examined the effects of video intervention on prevention of gestational diabetes mellitus (GDM). The authors provided a video containing advices about mediterranean diet and physical activity. The results showed the effectiveness. Although this manuscript has important results, there are some comments.
- The authors did not show maternal weight gain during pregnancy. I think that this is informative for readers.
We appreciate the comment. We have added a variable of the average gestational weight gain from pregestational BW to baseline and 24-28GW, in both groups to Table 2. It should be noted that no statistically significant differences were observed between groups.
Added data:
|
|
AT BASELINE |
24-28 GW |
Mean diferencies (95% CI) |
P value |
|
|
Maternal weight gain (Pregestational TO (Kg)) |
Control Group |
1.0 + 2.4 |
5.4 + 3.1 |
6.4 (-0.57; 8.77) |
0.001 |
|
Experimental Group |
1.2 + 2.7 |
4.9 + 2.3 |
6.2 (0.73; 7.65) |
0.001 |
|
|
P value |
0.186 |
0.444 |
0.081 |
|
|
- As a limitation, the authors mentioned the exclusion of some participants due to technical difficulties in downloading the video. I wonder if it caused some bias and made the results difficult to be interpreted. Please discuss it in more depth.
We have further developed the discussion to address this issue ((lines 339-346), analyzing the possible exclusion bias and explaining that, as this is a non-differential limitation (due to the technical incompatibility of devices), no substantial impact on internal validity is expected.
Added text (lines 350–357):
“Technical difficulties that prevented the video from being downloaded affected 10.45% of the women assigned to the experimental group. This potential bias can be considered non-differential, as it was not related to documented clinical or socioeconomic characteristics, but rather to the limitations of electronic devices. However, we acknowledge that it may have underestimated the actual effect of the intervention by excluding women with lower digital literacy. In future implementations, we propose facilitating access through viewing in waiting rooms or multiplatform formats to address this issue and overcome this limitation."
- I am interested in the effects of extending this method to the whole of Spain. I also recommend to consider the problems involved in expanding this throughout the country. Please discuss it.
We have edited the corresponding paragraph in the Discussion (lines 317-328), to address the feasibility of its national expansion and the possible logistical and sociocultural limitations.
Added text (lines 328-339):
“A major strength of this intervention lies in its scalability, simplicity, and feasibility for large-scale implementation. The video developed can be disseminated universally at minimal additional cost, integrated into routine prenatal care, and standardized across diverse clinical settings. Compared with other digital modalities, such as mobile applications or continuous remote monitoring, video-based education requires minimal technological literacy, ensures message standardization, and can be implemented even in resource-limited contexts. The expansion of this model at the national level could be easily integrated into public prenatal care programs due to its low cost, reproducibility, and open-access availability on free platforms, which facilitates its adoption. Nevertheless, effective implementation would require ensuring digital accessibility, particularly in rural areas, and fostering the active participation of obstetricians, midwives, and other healthcare professionals as educational reinforcement agents.”
- Abbreviations should be spelled out at the first time.
The entire manuscript has been reviewed, and all abbreviations have been expanded the first time they appear (e.g., “MEDAS: Mediterranean Diet Adherence Screener” “IADPSG: International Association of Diabetes and Pregnancy Study Groups” “WHO: World Health Organization” “CONSORT: Consolidated Standards of Reporting Trials”).
- There are some extra spaces in the main text.
A complete review of the format has been carried out and any spaces and typographical errors detected have been corrected.
- Table 1: Were history of GDM and miscarriage tested simultaneously? P value was shown as 0.004.
Thank you for pointing that out. The variables “history of GDM” and “previous abortion” were analysed by Chi2 test as Obstetric history. The table has been corrected as analysed these variables independently (Yes/no) and clarified in the table 1.

Reviewer 2 Report
Comments and Suggestions for Authors
Thank you for the article, it is quite interesting. However, there are some major and minor improvements that have to be made:
-
Clarify the randomization procedure: The allocation by weekday (3 vs 2 days) could introduce selection bias. Please explain how this method ensured true randomization and whether investigators were blinded during recruitment and data analysis.
-
Address baseline imbalances: Table 1 shows significant differences in systolic and diastolic blood pressure and ethnicity between groups (p<0.01). Discuss whether these differences could have influenced outcomes (e.g., hypertension or GDM rates) and consider adjusting for them in a multivariable model.
-
Expand on adherence assessment: Define how “viewing the video” was verified (self-report vs electronic tracking) and clarify how adherence influenced the per-protocol analysis. Quantify how many participants actually watched the video at least once and whether “non-watchers” were included in intention-to-treat analysis.
-
Statistical analysis refinement: Please specify whether adjustments were made for confounders (BMI, age, parity, ethnicity). A multivariate logistic regression for GDM incidence would strengthen causal interpretation and would bring value to the research.
-
Interpret clinical significance: The absolute reduction in GDM (5%) is statistically significant but modest. Discuss its clinical relevance in the context of population-level impact and cost-effectiveness, possibly quantifying the number needed to treat (NNT).
-
Digital accessibility and feasibility : Around 10% of women could not access the video. Please discuss the potential socioeconomic or digital literacy barriers and how they might affect the scalability of such interventions.
-
Clarify video content availability : The abstract and Data Availability Statement mention the video on Zenodo—please specify language(s), duration, and whether subtitles or multilingual versions are planned.
And also some minor things:
-
-
Correct minor typographical errors (e.g., “vied it more than once” → “viewed”).
-
Ensure consistent abbreviation use (e.g., GW, GDM).
-
Correct formatting issues in Tables 1–3.
-
-
Add participant flowchart details.
Figure 1 should explicitly show numbers excluded, lost to follow-up, and analyzed per CONSORT 2025 checklist. -
Discussion refinement.
Streamline paragraphs to avoid redundancy (lines 270–286). Strengthen the comparison with other tele-education studies and emphasize that such tools should complement, not replace, professional nutritionist support.
Author Response
Thank you for the article, it is quite interesting. However, there are some major and minor improvements that have to be made:
Comments and Suggestions for Authors
Thank you for the article, it is quite interesting. However, there are some major and minor improvements that have to be made:
- Clarify the randomization procedure: The allocation by weekday (3 vs 2 days) could introduce selection bias. Please explain how this method ensured true randomization and whether investigators were blinded during recruitment and data analysis.
We appreciate the observation. We have rewritten the paragraph referring to Randomization and Allocation to clarify this issue (lines 98- 116).
“Randomization was stratified by weekday clinics, alternating schedule designed to ensure a pragmatic yet unbiased allocation of participants. Specifically, for one week, women attending prenatal visits on three predetermined weekdays (Monday, Wednesday, Friday) were assigned to the experimental group, while those attending on the remaining two days (Tuesday, Thursday) were assigned to the control group. The following week, the allocation was reversed (the same weekdays were assigned to the control group and the others to the intervention group). This alternating assignment by working days was implemented to ensure pragmatic randomization in the consecutive flow of patients, avoiding any subjective selection by researchers. This procedure maintained random patient flow within each clinic day and avoided any subjective selection by investigators. Allocation was implemented by obstetricians during the first prenatal visit (<16 GW) by providing a unique Quick-Response (QR) code to experimental group participants. It should be noted that a considerable number of women in the experimental group did not download the video, since randomization did not account for whether participants had mobile devices or applications compatible with QR code scanning. This design, however, avoided the potential bias of selecting women with greater access to digital technologies. Data analysts remained blind to group allocation throughout the statistical analysis.”
- Address baseline imbalances: Table 1 shows significant differences in systolic and diastolic blood pressure and ethnicity between groups (p<0.01). Discuss whether these differences could have influenced outcomes (e.g., hypertension or GDM rates) and consider adjusting for them in a multivariable model.
We appreciate the observation. The difference in ethnicity reflects the actual composition of our population, and the length of residence in Spain was recorded to contextualize dietary acculturation. Several studies have shown that the Mediterranean diet confers similar benefits in women from different ethnic groups, with no interaction by race (1,2). Regarding differences in baseline blood pressure, no higher rates of preeclampsia or gestational hypertension were observed in the control group, reducing the likelihood of confounding. Furthermore, differences in sBP and dBP are small and opposite in each group, so the difference in mean BP is not significant. We believe that a difference in BW of less than 200g between groups has no clinical relevance.
We have added a paragraph (lines 237-244) to explain these differences:
“Although statistically significant differences were observed in ethnicity and blood pressure at baseline, these reflected the real demographic composition of our catchment population rather than selection bias. Furthermore, differences in sBP and dBP are opposite in each group, so the difference in mean BP is not significant. Moreover, participants were asked about their length of residence in Spain to account for acculturation effects, and the protective impact of Mediterranean diet adherence has been consistently demonstrated across diverse ethnic groups.”
- Expand on adherence assessment: Define how “viewing the video” was verified (self-report vs electronic tracking) and clarify how adherence influenced the per-protocol analysis. Quantify how many participants actually watched the video at least once and whether “non-watchers” were included in intention-to-treat analysis.
Thank you for the comment. The “Methods” section (2.4 Intervention and follow-up) has been revised to further specify how viewing was verified through supervised self-reports and their inclusion in the per-protocol analysis (lines 157-158).
Added text (lines 157-158):
“Verification was performed by the nutritionist through structured questioning during the follow-up assessment.”
- Statistical analysis refinement: Please specify whether adjustments were made for confounders (BMI, age, parity, ethnicity). A multivariate logistic regression for GDM incidence would strengthen causal interpretation and would bring value to the research.
This aspect is detailed in the new paragraph in the Statistical Analysis section (lines 228–231):
Added text (lines 228–231):
“Multivariate logistic regression analyses were performed adjusting for pre-pregnancy BMI, and ethnicity for the primary outcomes. The results remained statistically significant after adjustment.” Ods R (95% CI) for GDM 0.78 (0.61-0.99) reference Control Group The results remained statistically significant after adjustment.
- Interpret clinical significance: The absolute reduction in GDM (5%) is statistically significant but modest. Discuss its clinical relevance in the context of population-level impact and cost-effectiveness, possibly quantifying the number needed to treat (NNT).
Thank the reviewer for this comment. Although the absolute risk reduction (ARR) allows, in theory, the estimation of the number needed to treat (NNT) and its confidence interval, we believe that this parameter would not be fully interpretable in the present context. Our intervention (a short educational video) was not designed to replace the standard in-person nutritional counseling, but to reinforce adherence to the Mediterranean Diet pattern during pregnancy. Therefore, reporting the ARR and relative risk reduction provides a clearer and more accurate description of the intervention’s impact.
From an economic perspective, the prevention of a single case of GDM may represent an estimated short-term cost saving of around €818 per pregnancy in European settings (3).
Added text (lines 287–302):
“The absolute reduction of 5% in the incidence of GDM implies an estimated number needed to treat (NNT) of 22,7 women to prevent one case, which represents a clinically relevant impact at the population level given the simplicity and low cost of the intervention. If universally applied, approximately 1 in 23 cases of GDM could be prevented, with positive repercussions on maternal-fetal morbidity and healthcare system costs. Although the absolute risk reduction (ARR) observed in our study could theoretically be used to estimate the NNT and its confidence interval, this parameter might not be appropriate for direct interpretation, as the intervention was not intended to replace standard face-to-face dietary counseling but to enhance adherence to the Mediterranean Diet during pregnancy. Therefore, the ARR and relative risk reduction were considered more meaningful indicators of clinical relevance.
From a cost-effectiveness perspective, previous studies have estimated that pre-venting one case of GDM could save approximately 818 € per pregnancy in short-term healthcare costs suggesting that even modest reductions in incidence may have a considerable impact when extrapolated to large populations.”
- Digital accessibility and feasibility : Around 10% of women could not access the video. Please discuss the potential socioeconomic or digital literacy barriers and how they might affect the scalability of such interventions.
This aspect has been integrated together with the limitation already discussed in lines 350–357, where the need for inclusive strategies and alternative formats (screens in consultations, open web access, mobile versions) is emphasized.
Added text (lines 350–357):
“Technical difficulties that prevented the video from being downloaded affected 10.45% of the women assigned to the experimental group. This potential bias can be considered non-differential, as it was not related to documented clinical or socioeconomic characteristics, but rather to the limitations of electronic devices. However, we acknowledge that it may have underestimated the actual effect of the intervention by excluding women with lower digital literacy. In future implementations, we propose facilitating access through viewing in waiting rooms or multiplatform formats to address this issue and overcome this limitation.”
- Clarify video content availability : The abstract and Data Availability Statement mention the video on Zenodo—please specify language(s), duration, and whether subtitles or multilingual versions are planned.
Details about the video have been added to the intervention and follow-up section (lines 122-125).
Added text (lines 122-125)
“The video, recorded in Spanish and including Spanish subtitles, is freely available on the YouTube platform and designed to ensure universal accessibility. Translation into other languages, including English, is planned to facilitate future multicenter implementation.”
And also some minor things:
- Correct minor typographical errors (e.g., “vied it more than once” → “viewed”).
We apologize for the typographical errors. The whole text has been thoroughly reviewed.
- Ensure consistent abbreviation use (e.g., GW, GDM).
We have revised all the abbreviations and the first time they appear to ensure correct explanations and use.
- Correct formatting issues in Tables 1–3.
We have revised the format of all tables.
- Add participant flowchart details.
Figure 1 should explicitly show numbers excluded, lost to follow-up, and analyzed per CONSORT 2025 checklist.
We appreciate this suggestion. Following the reviewer's suggestion, we have replaced the flow chart
- Discussion refinement.
Streamline paragraphs to avoid redundancy (lines 270–286). Strengthen the comparison with other tele-education studies and emphasize that such tools should complement, not replace, professional nutritionist support.
Following the reviewer's suggestion, the relevant paragraphs have been reworked.
- Minhas AS, Hong X, Wang G, Rhee DK, Liu T, Zhang M, et al. Mediterranean-Style Diet and Risk of Preeclampsia by Race in the Boston Birth Cohort. J Am Heart Assoc. 3 de mayo de 2022;11(9):e022589.
- Makarem N, Chau K, Miller EC, Gyamfi-Bannerman C, Tous I, Booker W, et al. Association of a Mediterranean Diet Pattern With Adverse Pregnancy Outcomes Among US Women. JAMA Netw Open. 22 de diciembre de 2022;5(12):e2248165.
- Meregaglia M, Dainelli L, Banks H, Benedetto C, Detzel P, Fattore G. The short-term economic burden of gestational diabetes mellitus in Italy. BMC Pregnancy Childbirth. 23 de febrero de 2018;18(1):58.

Round 2
Reviewer 1 Report
Comments and Suggestions for Authors
To the authors
All of my criticisms have been addressed.
Reviewer 2 Report
Comments and Suggestions for Authors
Thank you for taking into consideration all our remarques.